# The quorum sensing transcription factor AphA directly regulates natural competence in *Vibrio cholerae*

**James R. J. Haycocks**[1], **Gemma Z. L. Warren**[1], **Lucas M. Walker**[1], **Jennifer L. Chlebek**[2], **Triana N. Dalia**[2], **Ankur B. Dalia**[2], **David C. Grainger**[1]*

**1** Institute of Microbiology and Infection, School of Biosciences, University of Birmingham, Edgbaston, Birmingham, United Kingdom, **2** Department of Biology, Indiana University, Bloomington, IN, United States of America

* d.grainger@bham.ac.uk

**Data Availability Statement:** Sequencing data have been deposited in ArrayExpress under accession code E-MTAB-7953. All other data

## Abstract

Many bacteria use population density to control gene expression via quorum sensing. In *Vibrio cholerae*, quorum sensing coordinates virulence, biofilm formation, and DNA uptake by natural competence. The transcription factors AphA and HapR, expressed at low and high cell density respectively, play a key role. In particular, AphA triggers the entire virulence cascade upon host colonisation. In this work we have mapped genome-wide DNA binding by AphA. We show that AphA is versatile, exhibiting distinct modes of DNA binding and promoter regulation. Unexpectedly, whilst HapR is known to induce natural competence, we demonstrate that AphA also intervenes. Most notably, AphA is a direct repressor of *tfoX*, the master activator of competence. Hence, production of AphA markedly suppressed DNA uptake; an effect largely circumvented by ectopic expression of *tfoX*. Our observations suggest dual regulation of competence. At low cell density AphA is a master repressor whilst HapR activates the process at high cell density. Thus, we provide deep mechanistic insight into the role of AphA and highlight how *V. cholerae* utilises this regulator for diverse purposes.

## Author summary

Cholera remains a devastating diarrhoeal disease responsible for millions of cases, thousands of deaths, and a $3 billion financial burden every year. Although notorious for causing human disease, the microorganism responsible for cholera is predominantly a resident of aquatic environments. Here, the organism survives in densely packed communities on the surfaces of crustaceans. Remarkably, in this situation, the microbe can feast on neighbouring cells and acquire their DNA. This provides a useful food source and an opportunity to obtain new genetic information. In this paper, we have investigated how acquisition of DNA from the local environment is regulated. We show that a "switch" within the microbial cell, known to activate disease processes in the human host, also controls DNA uptake. Our results explain why DNA scavenging only occurs in suitable

generated or analysed during this study are
included in the manuscript and supporting files.

**Funding:** This work was supported the
Biotechnology and Biological Sciences Research
Council (BBSRC, https://bbsrc.ukri.org/) with grant
BB/N005961/1 awarded to DCG and a PhD
studentship awarded to LMW. The National
Institutes of Health (NIH, https://grants.nih.gov/)
supported ABD with grant R35GM128674. The
funders had no role in study design, data collection
and analysis, decision to publish, or preparation of
the manuscript.

**Competing interests:** The authors have declared
that no competing interests exist.

environments and illustrates how interactions between common regulatory switches affords precise control of microbial behaviours.

## Introduction

*Vibrio cholerae* colonises two distinct habitats; the human intestine and aquatic ecosystems [1]. In the aquatic niche, the microbe forms biofilms on chitinous surfaces [2,3]. This induces expression of the gene regulatory protein TfoX [4–7]. Natural competence, the ability to acquire exogenous DNA from the environment, is triggered as a result [8,9]. Specifically, TfoX stimulates production of a type IV pilus that extends to bind, and retracts to internalise, exogenous DNA [10–13]. Genes encoding important cell envelope factors ComEA and ComEC, are also TfoX regulated [14]. Biofilms disperse upon entering the human gut [1,15]. This triggers the expression of virulence factors. Notably, the toxin co-regulated pilus (TCP) and cholera toxin (CT) are produced [16]. These factors are encoded by the *tcpPH* and *ctxAB* operons respectively.

Quorum sensing detects changes in bacterial population density reported by auto-inducer molecules [17]. This information is used to modify patterns of gene expression [18]. For example, CqsS is a membrane bound sensor kinase that detects cholera auto-inducer 1 (CAI-1) [19]. When kin are scarce, and CAI-1 levels low, CqsS triggers a regulatory cascade, which culminates in expression of five quorum regulatory RNAs [20]. These RNA molecules activate translation of AphA, a PadR family transcription factor with an N-terminal winged helix-turn-helix DNA binding motif [20–22]. In turn, AphA activates expression of *tcpPH* [23]. This event ignites the entire virulence gene expression programme [16]. Surprisingly, given the central role of AphA, the regulator is poorly understood. For instance, transcriptome analysis found just 6 operons controlled by AphA in *V. cholerae*, 5 of these indirectly [24]. Conversely, in closely related *Vibrio harveyi*, perturbation of *aphA* impacted transcription of hundreds of genes [22,25]. The DNA binding properties of AphA are also incompletely defined. Three studies have proposed a DNA consensus motif that binds AphA [24,26,27]. Though there are similarities, the sequence differs in each report. Thus, whilst a key factor in *V. cholerae*, the extent and mechanistic basis of regulation by AphA is not understood.

In this study, we sought to better understand the DNA binding and gene regulatory properties of AphA in *V. cholerae*. Using chromatin immunoprecipitation (ChIP) and DNA sequencing (ChIP-seq) we mapped genome-wide DNA binding by AphA. This revealed a palindromic consensus for AphA recognition. Interactions can involve single targets or co-operative AphA binding to adjacent sites. Though discovered as an activator of pathogenicity, AphA mainly targets genes related to cell surface physiology, including a subset required for natural competence. These target genes encode components of the type IV pilus, the competence regulator TfoX, and the quorum sensing factor CqsS. The mechanistic details vary but, at each locus, AphA acts as a transcriptional repressor. We conclude that AphA plays a key role by coupling natural competence to population density in *V. cholerae*.

## Results

### Genome-wide distribution of AphA in *Vibrio cholerae*

To identify genes targeted by AphA we used ChIP-seq. The data are illustrated in Fig 1A. In each plot, genes are shown as blue lines (outer two tracks) and the binding profile of AphA is in teal (inner track). We identified 49 binding peaks for AphA. The peaks were not distributed

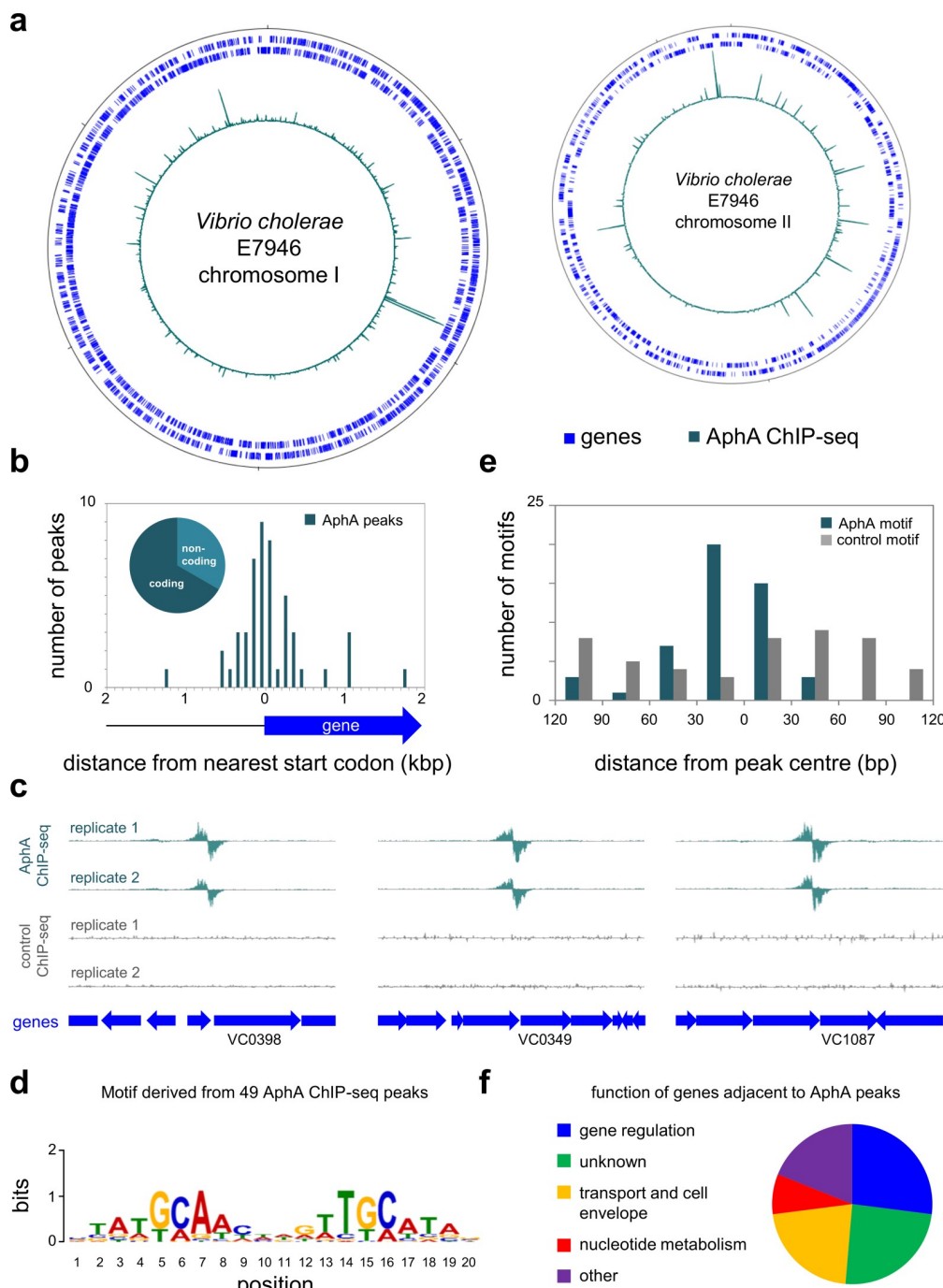

**Fig 1. Genome-wide binding of AphA in *Vibrio cholerae*.** (a) Binding of AphA across both *Vibrio cholerae* chromosomes. In each plot the outer two tracks (blue) are genes orientated in the forward or reverse direction. The AphA ChIP-seq binding signal (teal) is the inner profile. Position 1 of each chromosome is denoted by the tick mark at the top of each plot. Further tick marks are 1 Mbp apart. (b) Position of AphA binding peaks with respect to genes. The histogram shows the distribution of AphA binding peak centres with respect to the start codon of the nearest gene. The pie chart shows the proportion of binding peaks in coding or non-coding DNA. (c) Example AphA ChIP-seq binding peaks. The panel illustrates intragenic AphA binding peaks close to gene (block arrow) start codons. ChIP-seq coverage plots are shown for individual experimental replicates. Data for AphA are in teal and control profiles are grey. Signals above or below the horizontal line correspond to reads mapping to the top or bottom strand respectively. (d) Sequence motif derived from AphA binding peaks using MEME. (e) Distances between AphA binding peak centres and motifs. Data are shown as a histogram. The AphA motif is that shown in panel d. The control motif (5'-NAANTGCANTTN-3') was identified by MEME as lacking statistical significance (E = 1.1e+25). (f) Pie chart showing gene classes targeted by AphA.

evenly; 27% were located within a 0.5 Mbp section of chromosome II accounting for only 12% of the total genome (Fig 1A, right arm of chromosome II). To assess data validity, we determined the position of each peak with respect to the nearest gene start codon (Fig 1B). As expected, AphA peak positions cluster at the 5' ends of genes. Counterintuitively, this was true even though two thirds of peaks were located within coding sequences (Fig 1B, inset). This apparent contradiction arises because many peaks close to start codons are within adjacent genes. Examples of such peaks are shown in Fig 1C. Next, we extracted DNA motifs from the peak sequences using MEME. A single significant ($E = 2.2^{-7}$) DNA motif was found (Fig 1D). This palindromic sequence (5'-ATGCAA-$N_4$-TTGCAT-3') likely represents the preferred AphA binding target. We determined the distance between each occurrence of the motif and the centre of the ChIP-seq peak. As expected, the motif is biased to the centre of the ChIP-seq signal (Fig 1E, teal bars). The MEME analysis also identified motifs that were not statistically significant. These sequences were randomly distributed with respect to AphA peak centres. An example is shown as a control in Fig 1E (grey bars). The position of each peak, adjacent genes, and AphA binding motif is listed in Table 1. The distribution of functions associated with AphA targeted genes is shown in Fig 1F. We found that AphA primarily targeted genes for other regulatory proteins and components of the cell envelope. Most interestingly, AphA bound at three loci adjacent to genes known to influence DNA uptake by natural competence (Table 1). Briefly, these loci are the intergenic region between VC0857 and VC0858, the VC1153 (*tfoX*) promoter region, and the regulatory DNA for VCA0522 (*cqsS*). Importantly, purified AphA also recognised these targets specifically *in vitro* (S1 Fig). Hence, we next sought to understand the role of AphA at each locus.

## AphA binds a single site at the intergenic region between VC0857 and VC0858

The genes VC0857 and VC0858 encode homologues of the minor pilins *pilE* and *fimT*. Such proteins are hypothesised to initiate assembly of the DNA uptake pilus and reside at the tip of the pilus fibre [28,29]. Indeed, the VC0858 and VC0857 gene products contribute to pilus-DNA interactions, which occur specifically at the pilus tip [13]. The ChIP-seq data for AphA binding at the intergenic region between VC0857 and VC0858 is shown in Fig 2A. The sequence of the intergenic region is shown in Fig 2B. To precisely identify the AphA binding site we used DNAse I footprinting (Fig 2C). The data show that AphA binding protects a 25 bp region of DNA from digestion by DNAse I. The footprint coincides with a DNA sequence (ATGAAT-$N_4$-TTGCAT) that matches our motif for AphA binding at 10 of 12 positions (boxed in Fig 2B). The sequence also coincides precisely with the centre of the ChIP-seq peak (asterisk in Fig 2B).

## AphA prevents binding of RNA polymerase to the VC0857/VC0858 intergenic region and inhibits transcription

The previously defined transcription start sites (+1) for VC0857 and VC0858 are denoted by bent arrows in Fig 2B [30]. The corresponding promoter DNA elements are underlined. The AphA binding site overlaps the transcription start site for VC0858 and the promoter -35 element for VC0857 (Fig 2B). We reasoned that binding of AphA would be incompatible with binding of RNA polymerase to the regulatory region. To test this, we used electrophoretic mobility shift assays (EMSAs). The data show that AphA retarded migration of the regulatory DNA fragment (Fig 2D, compare lanes 1 and 2). A substantial change in electrophoretic mobility was evident when RNA polymerase was added instead of AphA (Fig 2D, compare lanes 1 to 3). Inclusion of AphA in incubations with RNA polymerase resulted in DNA

**Table 1. Position and sequence of AphA binding sites across the *V. cholerae* genome.**

| Peak Centre[1] | Motif centre[2] | Motif sequence (5' to 3')[3] | Gene(s)[4] | Adjacent gene(s)[5] |
|---|---|---|---|---|
| | | *Chromosome I* | | |
| 99851 | 99843 | ctatgaaattaaattaataa | VC0102[a] | VC0102[a] |
| 331346 | 331333 | gtagtcgagcagttgcttac | (VC0317) | |
| 371706 | 371662 | ggatgcaacaggtcgattgg | (VC0348) | VC0349 |
| 423066 | 423063 | ctatgcaacaatctgcgcct | (VC0397) | VC0398 |
| 519136 | 519147 | caatgcaactgctctcttac | (VC0487) | |
| 707906 | 707923 | ctaagcagcgaaatgcatac | VC0661 < > VC0662[r] | VC0661 < > VC0662[r] |
| 924229 | 924216 | ctatgcaagttgattcatca | VC0857[a] < > VC0858 | VC0857[a] < > VC0858 |
| 929346 | 929341 | gcaggcgccctgttgcatag | (VC0864) | |
| 936166 | 936182 | ctattcaacaagtttccaca | (VC0870) | |
| 1046511 | 1046514 | cgcagcatccaattgcattg | (VC0982) | VC0982 |
| 1090236 | 1090207 | ctatgcaacataatcaatct | VC1020 < > VC1021 | VC1020 < > VC1021 |
| 1154126 | 1154085 | ctatgcaggtgattgcagaa | (VC1086)[a] | VC1087 |
| 1224784 | 1224685 | ccatgcattttaccgattac | VC1152 < > VC1153[a] | VC1152 < > VC1153[a] |
| 1638101 | 1637985 | caagtcgtagtgatgctcac | (VC1522) | VC1522 < > VC1523[a] |
| 1656711 | 1656747 | ctgagcatactgttgcatcc | (VC1543) | |
| 1882301 | 1882329 | ttgctcactgagtttcatac | (VC1746), (VC1747) | VC1746 < > VC1747 |
| 2234986 | 2234972 | ggttgcaacggtttgaatac | (VC2077) | |
| 2327841 | 2327831 | atatgaagctatttgctggt | (VC2183) | VC2183 |
| 2364316 | 2364236 | ccatgcgcttgattgcaagt | VC2211 < > VC2212[r] | VC2211 < > VC2212[r] |
| 2435536 | 2435532 | ctaagaagccgagtgcataa | VC2280 | VC2280 |
| 2466111 | 2466116 | agatgcaaattgttgcatct | VC2317 | VC2317 |
| 2568506 | 2568491 | gttttaaatctgttgcatac | (VC2400) | VC2339 |
| 2628331 | 2628303 | ctgtgaacctagacgcatgg | (VC2448) | VC2447 |
| 2690451 | 2690442 | accggcaacatgatgcagcc | (VC2503) | |
| 2762401 | 2762392 | acgagcaacaagttgcgcgt | (VC2593) | VC2592 |
| 2810891 | 2810831 | gcatgaaacgtattgctggc | (VC2638) | |
| 2821251 | 2821257 | gtattccactttatgcttat | VC2647[r] | VC2647[r] |
| 2926246 | 2926235 | tctctcaactcaatgcatac | (VC2750) | VC2751 |
| | | *Chromosome II* | | |
| 39905 | 39881 | cttatctacttgatgcatag | VCA0032[a]<>VCA0033 | VCA0032[a]< >VCA0033 |
| 71933 | 71948 | ggctgcatcagtttgcttac | (VCA0064)[r] | |
| 83056 | 83013 | gtaggaatttcatttcatac | (VCA0074)[r] | |
| 105925 | 105897 | gtatgaaaccttagtcatgg | (VCA0098) | |
| 180601 | 180618 | ctctgcatccagatgcagga | (VCA0162) | VCA0163 |
| 215501 | 215472 | tcattcaagcgtttgcatag | (VCA0198)[r] | VCA0199 |
| 220027 | 219980 | tgtggaaacttgttgaatag | (VCA0202) | |
| 270051 | 270033 | catttccacaggtttcataa | VCA0249[a] | VCA0249[a] |
| 292440 | 292420 | tgtggaaacttgttgaatag | (VCA0275) | |
| 310741 | 310644 | ctatggtttttttgcatag | VCA0291 | VCA0291 |
| 354757 | 354789 | ctatgcgctttttttgcttag | VCA0367[a] | VCA0367[a] |
| 428574 | 428600 | ctattcaacaagtttccaca | (VCA0493) | VCA0494 |
| 454719 | 454697 | atgtgcatgttgatgattag | VCA0522[a] | VCA0522[a] |
| 772158 | 772161 | caatgcaaccaaatgaacag | VCA0826 | VCA0826 |
| 802525 | 802499 | cgcatcaatcagttgcatct | (VCA0851) | |
| 897851 | 897797 | gcgtgcacaataattcatat | VCA0946[a] | VCA0946[a] |
| 908441 | 908327 | acgtgccaactttttcattag | VCA0959 | VCA0959 |
| 1035131 | 1035098 | gaattaaatcaatttcatat | (VCA1081) | |
| 1052121 | 1052131 | ctatccaacatgttgcgtat | (VCA1095)[a] | |

*(Continued)*

**Table 1.** (*Continued*)

| Peak Centre[1] | Motif centre[2] | Motif sequence (5' to 3')[3] | Gene(s)[4] | Adjacent gene(s)[5] |
|---|---|---|---|---|
| 1053076 | 1053158 | `atagtcaaccattttcctac` | (VCA1097) | VCA1097 < > VCA1098 |
| 1056176 | 1056200 | `ctactcaacatgttgctggc` | (VCA1100) | |

[1]The centre of the ChIP-seq peak was defined as the centre of the 300 bp region that passed our cut-off for peak selection. Co-ordinates are for the *Vibrio cholerae* N16961 genome.

[2]Centre of AphA binding site associated with each peak. Note that these are predictions generated by MEME.

[3]Sequence of the predicted AphA binding site generated by MEME.

[4]Indicates the position of the peak with respect to the nearest gene(s). If the gene name is bracketed then the peak falls within the gene. Otherwise the peak is upstream of the gene or between divergent (< >) genes. Those genes activated[(a)] or repressed[(r)] by the presence of quorum sensing molecules AI-2, DPO and CAI-1 are indicated [64]. Note that the auto-inducer molecules drive down levels of AphA and so have the opposite regulatory effect. Where annotated, the gene numbers correspond to *hflX* (VC0348), *mshH* (VC0398), *glmS* (VC0487), *brnQ* (VC0662), *pilE* (VC0857), *fimT* (VC0858), *tnpA* (VC0870, VCA0202, VCA0275 and VCA0493), *luxO* (VC1021), *tfoX* (VC1153), *feoB* (VC2077), *prsA* (VC2183), *viuA* (VC2211), *pyrG* (VC2448), *hutR* (VCA0064), *pncB* (VCA0098), *intI4* (VCA0291), *cqsS* (VCA0522), *malK* (VCA0946), *cheA-3* (VCA1095).

[5]This column indicates the gene(s) with the closest start codon to the peak. In many cases peaks within genes are close to a start codon of an adjacent gene. If no gene name is given then the peak centre is not close to a start codon. Those genes activated[(a)] or repressed[(r)] by the presence of quorum sensing molecules AI-2, DPO and CAI-1 are indicated [31]. Note that the auto-inducer molecules drive down levels of AphA and so have the opposite regulatory effect.

migration similar to that with AphA alone (compare lanes 2 and 4). Hence, AphA appears to interfere with the binding of RNA polymerase. To measure the impact of AphA on promoter activity we used *in vitro* transcription assays. Hence, the DNA sequence in Fig 2B was cloned, in either the forward or reverse orientation, upstream of the λ*oop* terminator in plasmid pSR. Transcripts generated from the cloned DNA can be quantified after electrophoresis. The 108 nt RNAI transcript derives from the pSR replication origin and serves as an internal control. We were unable to detect transcripts from the VC0858 promoter. Conversely, a transcript of the expected size was generated by the VC0857 promoter (Fig 2E, lane 1). Addition of AphA abolished production of the transcript but not synthesis of the RNAI control (Fig 2E, lane 2). We also fused the regulatory fragment to *lacZ* in plasmid pRW50T. Next, *V. cholerae* strains with or without *aphA* were transformed using the plasmid derivatives. As with our *in vitro* analysis, we detected no transcription from the *VC0858* promoter. Hence, we can infer little about potential regulation of *VC0858* by AphA. However, *lacZ* expression was driven by the VC0857 promoter. Recall that AphA is active at low bacterial population densities. Hence, we measured β-galactosidase activity in different phases of growth. The results are shown in Fig 2F. In wild type cells, VC0857 promoter activity increased rapidly once the culture reached the optical density of ~0.8. (Fig 2F, solid line). In cells lacking AphA, the promoter was active at a markedly lower population density (Fig 2F, dashed line). We conclude that AphA is a repressor of VC0857 expression and exerts this effect by occluding the binding site for RNA polymerase (Fig 2G). Interestingly, whilst VC0857 is activated by quorum sensing auto-inducer molecules, which prevent AphA expression, VC0858 is unresponsive (Table 1) [31].

## AphA co-operatively binds two adjacent sites at the *tfoX* promoter

We next turned our attention to the regulatory region upstream of the *tfoX* gene. Briefly, TfoX is a master activator of all genes required for natural competence [11]. Hence, the decision to express *tfoX* is a major checkpoint. The ChIP-seq data for AphA binding upstream of *tfoX* are shown in Fig 3A. The corresponding gene regulatory region is shown in Fig 3B. The centre of the AphA binding peak is again denoted by an asterisk. Note that expression of *tfoX* is almost completely dependent on CRP binding to a site (orange in Fig 3B) centred 83.5 bp upstream of

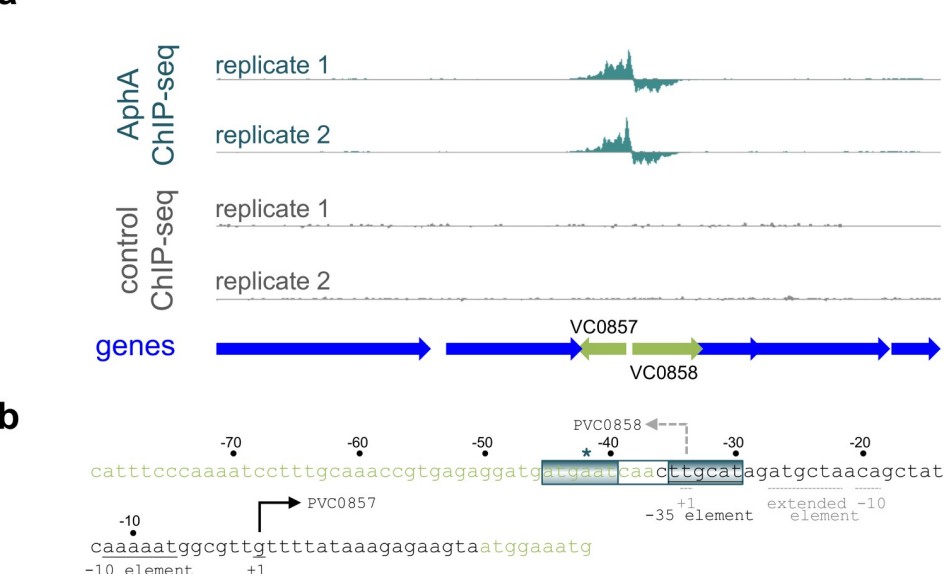

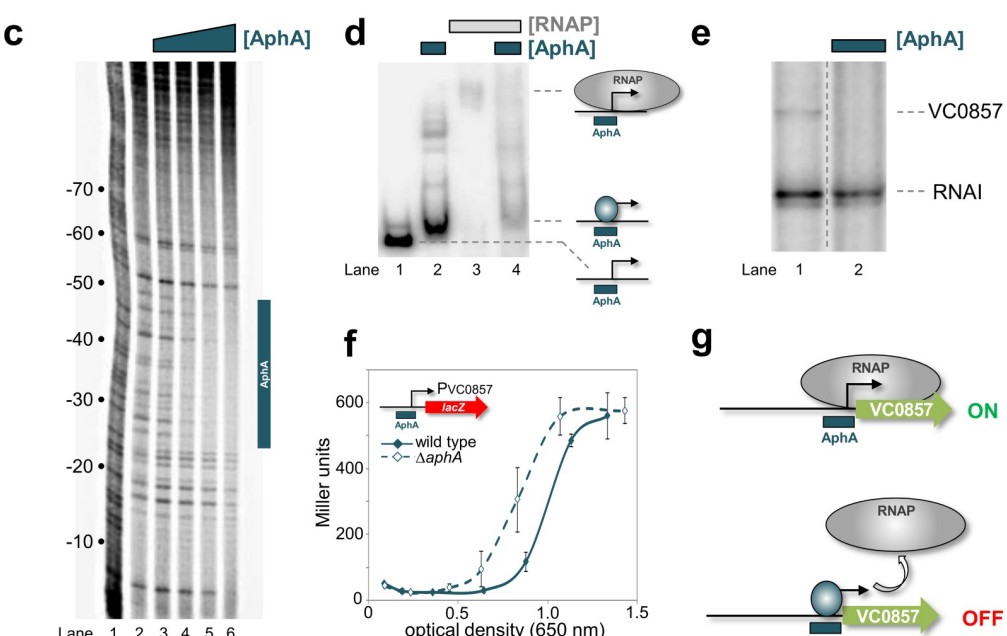

**Fig 2. Repression of VC0857 expression by AphA.** (a) AphA binding between VC0857 and V0858. Genes are shown as block arrows. ChIP-seq coverage plots are shown for individual experimental replicates. Data for AphA are in teal and control profiles are grey. Signals above or below the horizontal line correspond to reads mapping to the top or bottom strand respectively. (b) DNA sequence of the intergenic region between VC0857 and VC0858. For clarity, the sequence orientation has been inverted. Coding DNA sequence is light green. Bent arrows denote transcription start sites for VC0857 (solid line) and VC0858 (broken line). Promoter elements are labelled similarly. Distances (bp) are with respect to the VC0857 transcription start site. The teal asterisk denotes the ChIP-seq peak centre. The AphA binding site is boxed and labelled. (c) DNase I footprint of AphA bound at the intergenic region. Lane 1 is a Maxim-Gilbert 'G+A' ladder. Where added, the concentration of AphA was 0.25, 0.50, 1.00 or 2.00 μM. The footprint is indicated with a vertical teal bar. (d) AphA prevents RNA polymerase binding. Electrophoretic mobility shift assay showing mobility the VC0857-VC0857 intergenic DNA (lane 1), with 0.5 μM AphA (lane 2), 0.4 μM RNA polymerase (lane 3), or both factors (lane 4). (e) AphA represses VC0857 *in vitro*. The gel shows the result of an *in vitro* transcription experiment. Experiments were done with 0.4 μM RNA polymerase in the absence (lane 1) or presence (lane 2) of 1 μM AphA. The RNAI transcript acts as an internal control. The VC0857 transcript is labelled. No transcript was detected for the VC0858 promoter (see full gel image in S6 Fig, labelled red). (f) Activity of the VC0857

promoter *in vivo*. β-galactosidase activity was measured for *Vibrio cholerae* E7946 (solid teal line) or the Δ*aphA* derivative (broken teal line) containing the VC0857 promoter fused to *lacZ*. Ectopic expression of *tfoX* was provided by plasmid pMMB-*tfoX*. This is necessary because no VC0857 promoter activity is detectable in the absence of *tfoX*. Standard deviation is shown for three independent biological replicates. For the VC0858 promoter activity values were fewer than 10 Miller units. (g) Model for AphA-mediated repression at VC0857. AphA binds a single site overlapping the VC0857 promoter and occludes RNA polymerase binding.

the *tfoX* transcription start site (bent arrow in Fig 3B) [32]. We first used DNAse I footprinting to pinpoint AphA binding to the *tfoX* regulatory region (S2A Fig, lanes 2–4). The AphA footprint occurs between promoter positions -73 and -110. It is important to note that the upstream boundary of the footprint is demarked by a subtle change in the pattern of DNAse I digestion; protection of the base at position -109 and DNAse I hypersensitivity at position -110. This transition is marked by the teal arrow adjacent to lane 4 in S2A Fig. Since the AphA footprint extends over 37 bp it likely represents AphA binding two adjacent sites. The proposed sites are labelled AphA I and AphA II in Fig 3B. The sequences coincide precisely with the centre of the ChIP-seq peak for AphA binding. Sites AphA I (5'-CAA<u>CAA</u>-N$_4$-<u>T</u>TGACG-3') and AphA II (5'-G<u>TGATA</u>-N$_4$-<u>T</u>CT<u>CAT</u>-3') match the consensus for AphA binding at 6/12 and 7/12 positions respectively. Hence, these seem comparatively poor binding targets. To understand how AphA recognises these sequences we mutated site AphA I or AphA II (red in Fig 3B). We then used DNAse I footprinting to investigate the consequences. Mutations in the upstream AphA I site changed only the upstream half of the large AphA footprint (S2A Fig, lanes 5–7). Hence, the hypersensitive band at position -110 did not appear. We also observed poor protection of a band at position -100 (see triangle to the right of lane 7 in S2A Fig). Mutations in the downstream AphA II site had pronounced consequences. First, the mutations changed the overall pattern of DNAse I sensitivity (compare lanes 2 and 8 in S2A Fig). Second, the mutations rendered AphA unable to bind to either site I or site II (S2A Fig, lanes 8–10). We conclude that AphA binds its two adjacent sites at the *tfoX* regulatory region co-operatively. Hence, mutations in site AphA II, a closer match to the consensus, abolish recognition of both targets.

## AphA and CRP compete for overlapping binding sites upstream of the *tfoX* promoter

Strikingly, the region of the *tfoX* promoter bound by AphA overlaps the binding site for CRP (orange in Fig 3B). We reasoned that AphA and CRP may compete for binding. To test this, we further utilised DNAse I footprinting (Fig 3C). Fortuitously, although CRP and AphA bound similar locations, the footprints produced were easily distinguished. Thus, CRP binding resulted in DNAse I hypersensitivity at positions -83 and -92 upstream of the *tfoX* promoter (Fig 3C, lane 3). As already noted, AphA binding protected the DNA and induced DNAse I hypersensitivity at promoter position -110 (Fig 3C, lane 4). The sites of DNAse I hypersensitivity due to CRP and AphA binding respectively are shown by orange and teal triangles to the right of Fig 3C. Next, we added increasing concentrations of AphA to incubations containing the *tfoX* promoter DNA fragment and CRP (Fig 3C, lanes 5–9). As the concentration of AphA increased so did the occurrence of DNAse I hypersensitivity at position -110. Concomitantly, DNAse I hypersensitivity at positions -83 and -92, due to CRP binding, was reduced. In parallel experiments we measured binding of AphA and/or CRP to the *tfoX* promoter region using EMSAs (Fig 3D). Addition of AphA or CRP to incubations altered migration of the *tfoX* promoter DNA fragment during electrophoresis. Importantly, the degree to which migration altered was different for each protein (compare lanes 1–3 in Fig 3D). This is most likely because CRP bends the DNA by 90˚ whilst AphA has little effect [21,33,34]. Addition of AphA

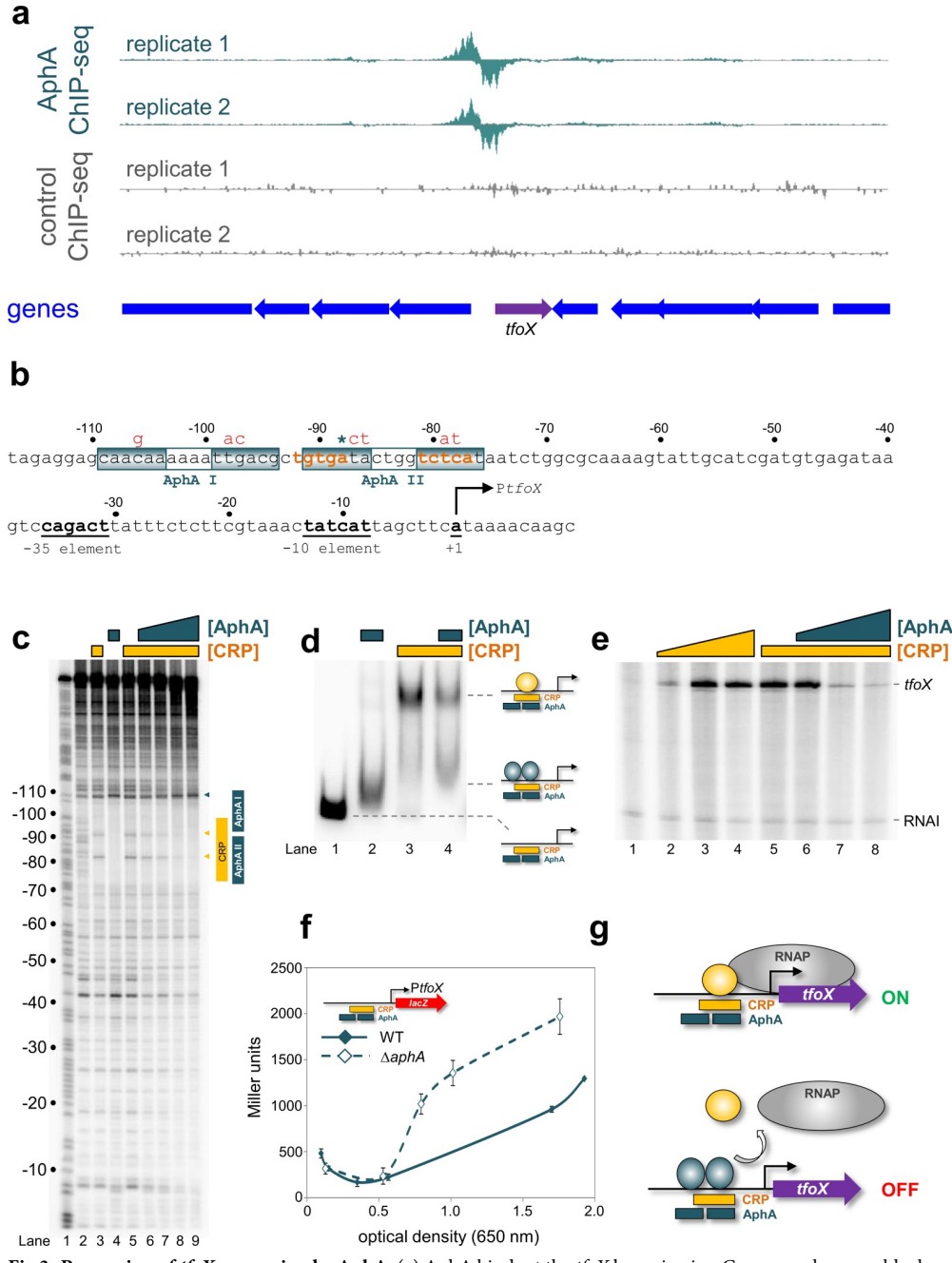

**Fig 3. Repression of *tfoX* expression by AphA.** (a) AphA binds at the *tfoX* locus *in vivo*. Genes are shown as block arrows. ChIP-seq coverage plots are shown for individual experimental replicates. Data for AphA are in teal and control profiles are grey. Signals above or below the horizontal line correspond to reads mapping to the top or bottom strand respectively. (b) DNA sequence of the *tfoX* regulatory region. The -35 and -10 elements are underlined and labelled, and the transcription start site is shown as a bent arrow. Distances are with respect to the *tfoX* transcription start site. The CRP binding site is orange and AphA sites are boxed. The ChIP-seq peak centre is denoted by an asterisk. Red bases above the sequence indicate mutations made to each AphA site. (c) Binding of AphA and CRP to overlapping sites at the *tfoX* promoter. The gel shows the result of DNase I footprinting experiment. The gel is calibrated with a Maxim/Gilbert 'G+A' ladder in lane 1. Protection of DNA from DNase I cleavage in the presence of 0.5 μM CRP is shown in lane 3 and highlighted by the vertical yellow bar. Sites of DNAse I hypersensitivity due to CRP binding are indicated by yellow triangles. Protection from DNase I cleavage in the presence of 1.0 μM AphA is shown in lane 4 and is highlighted by vertical teal bars. Increased DNase I cleavage induced by AphA is indicated by a teal triangle. In the presence of 0.5 μM CRP, increasing concentrations of AphA result in the replacement of the CRP footprint with an AphA footprint (lanes 5–9, AphA concentrations: 0, 0.5 μM, 1.0 μM, 2.0 μM, 3.0 μM). (d) AphA

outcompetes CRP for binding at the *tfoX* locus. Electrophoretic mobility shift assay showing the mobility of the *tfoX* promoter fragment alone (lane 1), in the presence of 0.5 μM AphA (lane 2), 1.0 μM CRP (lane 3), or both proteins (lane 4). (e) AphA antagonises CRP-dependent transcription activation at the *tfoX* promoter *in vitro*. The gel shows *tfoX* and RNAI (control) transcripts generated *in vitro*. Transcription from the *tfoX* promoter increased in the presence of CRP (lanes 2–4: 1, 3, 5 μM CRP). Addition of AphA with 3 μM CRP reduced *tfoX* transcription (lanes 5–8, concentrations of AphA: 0, 1, 3, 5 μM). (f) AphA regulates *tfoX* transcription *in vivo*. β-galactosidase activity was measured for *Vibrio cholerae* E7946 (solid teal line) or the Δ*aphA* derivative (broken teal line) containing the *tfoX* promoter fused to *lacZ*. Standard deviation is shown for three independent biological replicates. Cells were grown in LB medium. (g) Model for AphA repression of *tfoX*. CRP activates transcription from the *tfoX* promoter in the absence of AphA. At low cell density, AphA co-operatively binds two adjacent sites to displace CRP.

reduced the abundance of complexes due to CRP and increased the abundance of the complexes due to AphA (Fig 3D, lane 4). Thus, AphA and CRP compete for binding the same section of the *tfoX* gene regulatory region. Note that AphA could outcompete CRP even if the latter was present at higher concentrations (Fig 3D). We conclude that AphA is likely an anti-activator of *tfoX* expression (i.e. the repressor targets the activator rather than RNA polymerase directly).

## AphA prevents activation of *tfoX* expression mediated by CRP

To investigate the effects of AphA and CRP on *tfoX* promoter activity we used *in vitro* transcription assays. As expected, activation of *tfoX* transcription by CRP was evident (Fig 3E, lanes 1–4). Addition of AphA abolished activation by CRP (lanes 5–8). To investigate repression *in vivo* we utilised the *lacZ* reporter plasmid described above. In wild type cells *tfoX* promoter activity increased in line with culture optical density (Fig 3F, solid line). The pattern of *tfoX* promoter activity was different in the absence of AphA (Fig 3F, dashed line). In particular, β-galactosidase activity increased to 5-fold higher levels in the early- to mid-exponential phase of growth. Our model for regulation of *tfoX* promoter activity by AphA and CRP is shown in Fig 3G.

## AphA and CRP bind the *cqsS* regulatory region in unison

The final AphA target selected for characterisation was adjacent to *cqsS*. The CqsS protein is at the top of the regulatory cascade triggered by the quorum sensing auto-inducer molecule CAI-1 [20]. At high population densities the cascade prevents expression of downstream genes including *tfoX* and *aphA*. The ChIP-seq signal for AphA binding upstream of *cqsS* is shown in Fig 4A and the sequence of the regulatory region is shown in Fig 4B. We again used DNAse I foorprinting to dissect binding of AphA (S2B Fig). The footprint due to AphA was 37 bp in length (S2B Fig, lanes 2–4). This was indicative of two adjacent AphA sites (labelled I and II in Fig 4B). Mutations made in each site are shown in Fig 4B (red text) and footprints done using the mutated DNA fragments are in S2B Fig. Mutations in the upstream AphA I site completely abolished binding of AphA to both targets (S2B Fig, lanes 5–7) whilst mutations in the downstream AphA II site only prevented binding to the second occurrence of the motif (S2B Fig, lanes 8–10). Hence, we again conclude that AphA recognises two sites co-operatively at the *cqsS* promoter region. Importantly, the two sites overlap precisely with the DNAse I footprint and the centre of the ChIP-seq peak (asterisk in Fig 4B). The AphA sites also overlap the *cqsS* transcription start site (bent arrow in Fig 4B) [30]. We noticed that the *cqsS* regulatory region contained an 8/10 match to the consensus sequence for CRP binding (orange in Fig 4B) centred 41.5 bp upstream of the *cqsS* promoter. This is intriguing because CRP enhances expression of CqsA, which synthesises CAI-1 detected by CqsS [35]. Hence, we sought to understand if CRP bound this site. The results of DNAse I footprinting experiments are shown in Fig 4C.

As expected, CRP (lane 3) and AphA (lane 4) produced footprints at their respective target sites. Addition of AphA to incubations containing CRP resulted in footprinting of both the AphA and CRP targets (lanes 5–7). Similar results were obtained in parallel EMSAs (Fig 4D). Hence, AphA (lane 2) and CRP (lane 3) both individually bound to the *cqsS* DNA fragment. A super-shifted complex was observed when the proteins were added in unison (lane 4).

### AphA and CRP oppositely regulate the *cqsS* promoter

To understand the effects of AphA and CRP on *cqsS* promoter activity we first used *in vitro* transcription assays (Fig 4E). A transcript was only generated from the *cqsS* promoter in the presence of CRP, albeit at low levels (lanes 1 and 2). In the presence of CRP and AphA this transcript was undetectable (lanes 4–6). Because the *cqsS* promoter was poorly active in our *in vitro* transcription analysis we also used $KMnO_4$ footprinting. This detects DNA opening by RNA polymerase at promoter -10 elements during transcription initiation. The results are shown in Fig 4F; the appearance of bands is indicative of DNA melting. Such bands were only observed in the presence of both CRP and RNA polymerase (lane 6). Addition of AphA to incubations with CRP and RNA polymerase prevented promoter unwinding (lanes 7–9). The low level transcription detected *in vitro* (Fig 4E) was recapitulated using a *cqsS::lacZ* fusion *in vivo* (Fig 4G). Hence, only 120 Miller units were detected for experiments with the *cqsS* promoter; 20-fold lower than equivalent experiments with the *tfoX* promoter (compare Figs 3F and 4G). Surprisingly, β-galactosidase activity in lysates of wild type and Δ*aphA* cells were similar (compare solid and dashed lines in Fig 4G). We conclude that AphA is likely to be a repressor of the *cqsS* promoter. However, specific conditions may be required to detect such repression *in vivo* (Fig 4H). We note that addition of quorum sensing auto-inducer molecules, which block production of AphA, activate *cqsS* expression (Table 1) [31].

### Deleting *aphA* enhances natural competence at low cell density

Whether *V. cholerae* cells act individually or as a group is coupled to population density by the regulator LuxO [20,36]. Consequently, the LuxO[D61E] derivative (earlier misnamed LuxO[D47E]) locks cells in a low density state. This mutation is frequently exploited to study behaviours specific to this mode of life [22,37]. Our data suggest that AphA represses the regulatory cascade triggering natural competence at low population density. To test this, we introduced *luxO*[D61E] into *V. cholerae* E7946 and a derivative lacking *aphA*. We then measured the frequency of transformation by natural competence for each strain. The result of the experiment is shown in Fig 5A (horizontally lined bars). As expected, the E7946 strain encoding LuxO[D61E] was poorly transformable. Consistent with our model, deletion of *aphA* triggered a >500-fold increase in transformation frequency. We also measured the effect of deleting *dns*; a gene encoding an endonuclease expressed at low cell density to degrade any DNA obtained by natural transformation (compare striped and speckled bars) [38]. There was no significant effect of deleting *dns* when *aphA* was present. A 6-fold increase in transformation frequency was apparent when *dns* was lost from *aphA* null cells.

### Expression of *aphA* reduces natural competence at high cell density

We next examined transformation of E7946, and the derivative lacking *aphA*, in the context of the wild type *luxO* allele. These strains transition to a high cell density, and become naturally competent, upon colonisation of chitinous surfaces. In this scenario, wild type E7946 transformed efficiently. However, because *aphA* is not expressed at high cell density, deleting the gene had no effect (Fig 5A, open bars). We reasoned that differences in transformation frequency would be observed if *aphA* was expressed ectopically. To achieve this, *V. cholerae* strain

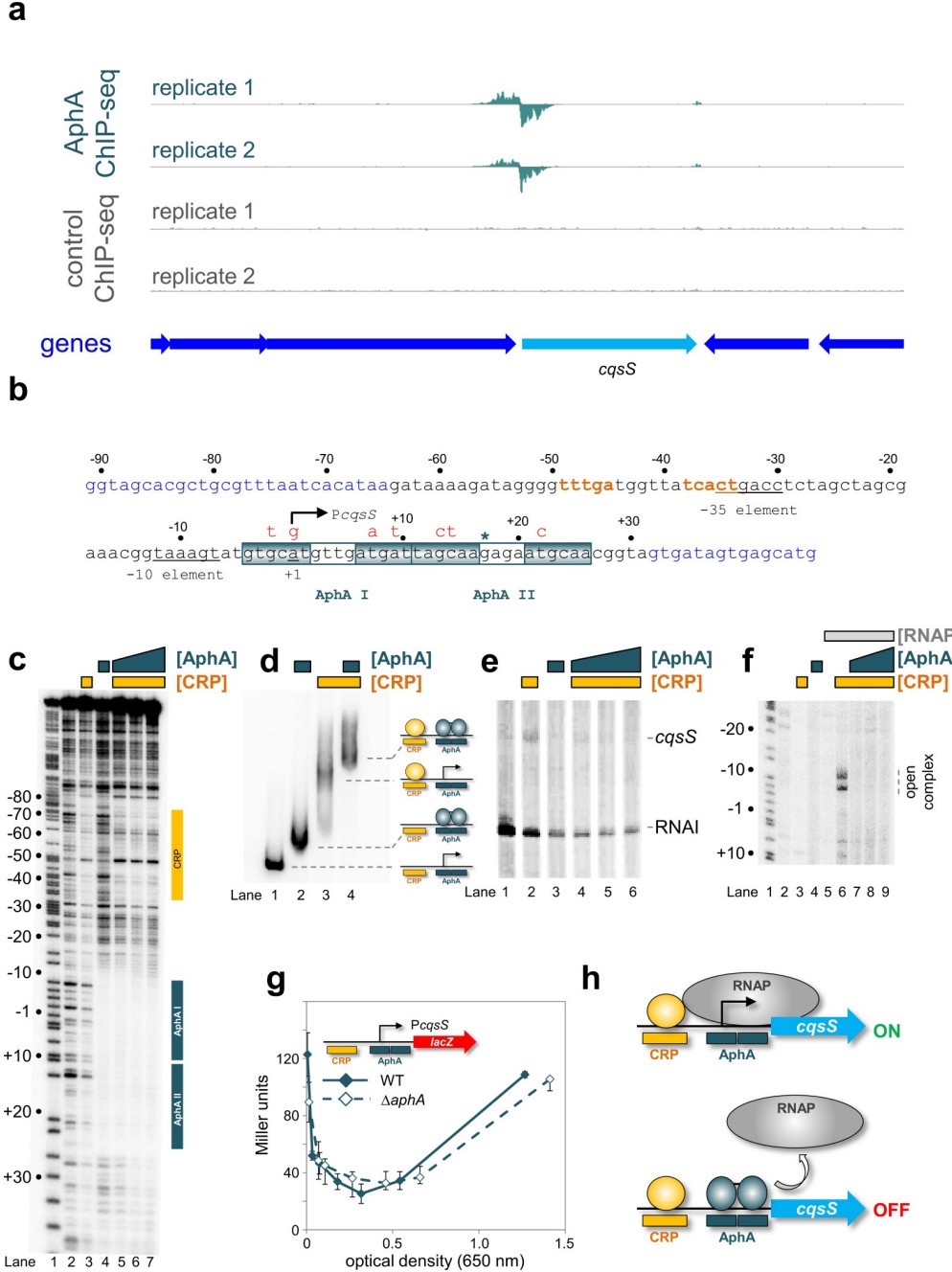

**Fig 4. Interactions between AphA, CRP and the *cqsS* regulatory region.** (a) AphA binds upstream of *cqsS in vivo*. Genes are shown as block arrows. ChIP-seq coverage plots are shown for individual experimental replicates. Data for AphA are in teal and control profiles are grey. Signals above or below the horizontal line correspond to reads mapping to the top or bottom strand respectively.(b) Sequence of the *cqsS* intergenic region. Coding DNA is in blue. Promoter elements are underlined and labelled. The transcription start site is shown as a bent arrow. Distances are with respect to the *cqsS* transcription start site. The CRP binding site is in orange and AphA binding sites are boxed. The centre of the AphA ChIP-seq peak is denoted by a teal asterisk. Mutations introduced into each AphA site are shown in red above the sequence.(c) AphA and CRP bind distinct sites at P*cqsS*. The gel shows the result of a DNase I footprint using a DNA fragment containing P*cqsS*. A Maxim-Gilbert 'G+A' ladder has been used to calibrate the gel (lane 1). The addition of 3 µM CRP produces a footprint indicated by the yellow vertical bar (lane 3). The vertical teal bar indicates the footprint formed in the presence of 1 µM AphA (lane 4). The CRP footprint is not altered by increasing concentrations of AphA (lanes 5–7, CRP concentration: 3 µM, AphA concentrations: 1, 2, 3 µM).(d) AphA and CRP bind to P*cqsS* simultaneously. The gel shows the result of electrophoretic mobility shift assay using a DNA fragment containing P*cqsS*. Data are shown for the DNA alone (lane 1), or in the presence of 1 µM AphA (lane 2), 3 µM CRP

(lane 3), or both proteins (lane 4). (e) AphA and CRP have antagonistic effects on P*cqsS in vitro*. Result of an *in vitro* transcription experiment using the P*cqsS* promoter cloned in pSR as a template. Reactions all contained 0.4 µM RNA polymerase. Transcription factors were added as follows: 3 µM CRP (lane 2), 1 µM AphA (lane 3) 3 µM CRP, and 1, 2, or 3 µM AphA (lanes 4–6). (f) Opening of the *cqsS* promoter by RNA polymerase is controlled by CRP and AphA. The gel shows the result of a KMnO$_4$ footprint. A Maxim/Gilbert 'G+A' reaction is shown in lane 1 for calibration. Lane 2 shows an absence of DNA opening without added proteins. Addition of 3 µM CRP (lane 3), 1 µM AphA (lane 4), or 300 nM RNA polymerase (lane 5) alone does not impact DNA melting. In the presence of 3 µM CRP and 300 nM RNA polymerase promoter unwinding was observed (lane 6). This was inhibited by AphA (lanes 7–9, AphA concentrations: 1, 2 and 3 µM). (g) Activity of P*cqsS in vivo*. β-galactosidase activity was measured for *Vibrio cholerae* E7946 (solid teal line) or the ∆*aphA* derivative (broken teal line) containing P*cqsS* fused to *lacZ*. Standard deviation is shown for three independent biological replicates. Cells were grown in M9 minimal media supplemented with 0.3% (*w/v*) fructose. (h) Model for AphA-mediated repression at P*cqsS in vitro*. CRP activates transcription from a site centred at position -41.5 relative to the *cqsS* transcription start site. AphA binds to two sites overlapping the transcription start site to repress transcription.

E7946 was transformed with plasmid pAMCF*aphA* that encodes the C-terminally 3xFLAG tagged AphA used in our ChIP-seq experiments. Importantly, the level of AphA generated from this plasmid precisely matches that of chromosomally encoded AphA (S3 Fig). As a control, we utilised plasmid pAMNF*aphA* encoding N-terminally 3xFLAG tagged AphA, which cannot bind DNA (see control ChIP-seq data). Expression of the active C-terminally tagged AphA reduced transformation frequency by ~1,300-fold. Comparatively, N-terminally tagged AphA had little effect (compare open and black bars in Fig 5A).

## Uncoupling TfoX expression from AphA regulation largely restores natural transformation

We next sought to understand which of the AphA regulatory events described above (Figs 2–4) resulted in the loss of competence phenotype due to constitutive AphA production. First, we focused our attention on *tfoX* repression by AphA. If responsible for reduced competence, natural transformation should be restored by uncoupling *tfoX* expression from AphA regulation. To do this we replaced the native *tfoX* promoter with the IPTG inducible *tac* promoter (P$_{tac}$-*tfoX*) in strains containing pAMCF*aphA* or pAMNF*aphA*. Expression of *tfoX* almost completely abolished the effect of constitutive AphA production on natural transformation (compare black and grey bars in Fig 5A). We conclude that AphA mediated *tfoX* repression is the primary cause of reduced natural transformation in our experiments.

## Constitutive expression of AphA does not impact pilus production

Our focus turned to the residual 6-fold effect of AphA observed in the presence of ectopic *tfoX* expression (see the difference between grey bars in Fig 5A). We reasoned that attenuated pilus activity, due to repression of VC0857 by AphA, might be responsible. To monitor pilus production, we utilised *V. cholerae* encoding PilA$^{S67C}$ [13]. The cysteine substitution facilitates *in vivo* labelling of PilA with the fluorescent dye AF488-mal. Since PilA is a major component of the DNA uptake apparatus, this allows visualisation of pili in live cells. Recall that TfoX is an activator of VC0857 [11]. To avoid indirect effects of AphA, mediated by repression of *tfoX*, we again utilised P$_{tac}$-*tfoX*. The supplementary S1 and S2 Movies, and the representative images in Fig 5B, show dynamic pilus events (white arrows). We compared cells expressing the non-functional N-terminally tagged AphA or the active C-terminally tagged derivative. A quantification of the data is shown in Fig 5C; there was no significant difference between strains (P = 0.06). We conclude that repression of VC0857 by AphA has little impact on DNA uptake in the conditions of our assay. This is consistent with previous reports of VC0857 having only minor effects on competence [10].

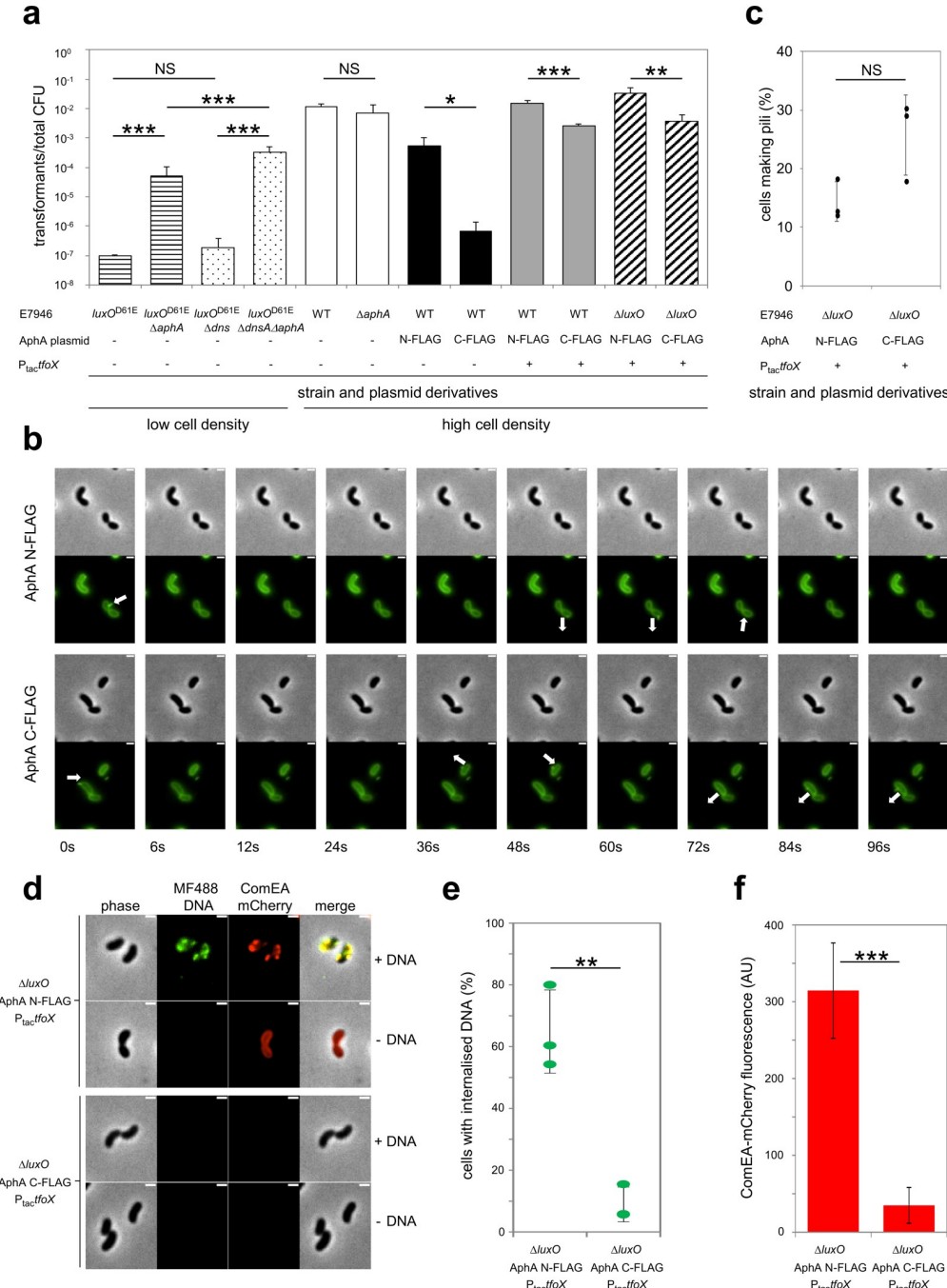

**Fig 5. AphA represses DNA uptake *in vivo* by repressing expression of TfoX and ComEA.** (a) Chitin-induced transformation assays at low or high cell density. The LuxO[D61E] derivative is constitutively active due to a phosphomimetic mutation that locks *V. cholerae* in a low cell density state. The *dns* gene encodes an endonuclease expressed at low (but not high) cell density to degrade DNA obtained by natural transformation. $P_{tac}$-*tfoX* is a chromosomally integrated IPTG-inducible construct. Transformation reactions for strains with $P_{tac}$-*tfoX* were supplemented with 100 μM IPTG. Plasmid encoded N- (N-FLAG) or C-terminally (C-FLAG) tagged AphA was constitutively expressed in strains as indicated. Transforming DNA was an antibiotic resistance cassette that integrates in place of VC1807 encoding Erm[R], Zeo[R] or Tet[R]. All data are from at least 4 independent biological replicates and P was calculated using a two-tailed Student's t-test. Data were significant at * P ≤ 0.05; ** P ≤ 0.01 or *** P ≤ 0.001. (b) Pilus production was assessed by timelapse microscopy using cells labelled with AF488-mal. Data are representative of at least 3 independent experiments. White arrows indicate dynamic pilus events and arrow orientation indicates extension or retraction. The scale bar is 1 μm. (c) Quantification of pilus dynamic activity (as shown in b) as the

percentage of cells that exhibited at least 1 dynamic pilus event. Data are from three independent experiments and n = 527 cells analysed for N-FLAG and n = 625 for C-FLAG. (d) Uptake of MFP488-labelled DNA was assessed in *comEA*-mCherrry, Ptac-*tfoX*, Δ*luxO* cells harbouring plasmids for constitutive expression of N- or C- terminally tagged AphA as indicated. Data are representative of at least 3 independent experiments. The scale bar is 1μm. (e) Quantification of DNA uptake (as shown in d) by assessing the percentage of cells that displayed periplasmically localized MFP488-labelled DNA. Data are from three independent experiments and n = 1227 cells analysed for N-FLAG and n = 1138 for C-FLAG. P = 0.002 (two tailed Student's t-test). (f) Quantification of ComEA-mCherry expression by measuring mCherry fluorescence in a microplate reader. Data are from four independent biological replicates. P = 0.0002 (two-tailed Student's t-test).

## Deleting *luxO* does not bypass inhibition of natural competence by AphA

To determine if repression of *cqsS* contributed to the remaining 6-fold drop in natural transformation, we deleted *luxO*. Deletion of *luxO* constitutively activates a high cell density expression profile regardless of CqsS activity or regulation. The 6-fold drop in natural transformation, caused by AphA in the presence of ectopic *tfoX* expression, was not altered by deleting *luxO* (compare grey and striped bars in Fig 5A).

## Constitutive expression of AphA reduces levels of ComEA

Repression of VC0857 or *cqsS* cannot explain the lingering impact of AphA when *tfoX* is ectopically expressed. We resolved to determine which aspect of natural transformation was still impaired. As a starting point we measured DNA uptake. Hence, we incubated P$_{tac}$-*tfoX* cells with MFP488-labeled DNA and counted cells with DNA in their periplasm. Bacteria that constitutively expressed the non-functional N-terminally tagged AphA acquired DNA efficiently. Conversely, cells expressing functional C-terminally tagged AphA showed a significant 6-fold reduction (Fig 5D and 5E). Hence, DNA uptake was reduced by AphA even when TfoX was produced. Acquisition of DNA results from the concerted activity of type IV pili and periplasmically localized ComEA, the latter of which acts as a molecular ratchet to facilitate DNA uptake [14,39,40]. As we had already determined that pilus production was not affected by AphA in the P$_{tac}$-*tfoX* background (Fig 5C), we hypothesized that AphA may impact ComEA production. Using a functional fluorescent fusion, we found that ComEA expression was significantly reduced by expression of C-terminally tagged AphA (Fig 5D and 5F). Since AphA does not bind upstream of *comEA* (S4 Fig) this repression must be indirect.

## Discussion

In *V. cholerae*, signals for natural competence are integrated at the *tfoX* promoter [4,5,14,32,41]. Triggers include: i) carbon starvation, mediated through CRP and ii) chitin-breakdown, sensed by membrane proteins ChiS and TfoS then communicated via the small RNA TfoR [5–8,32,42]. In comparison, regulatory links between cell density and competence are poorly characterised. At high cell density, HapR stimulates competence by blocking expression of the nuclease Dns [11,38]. Indirectly, HapR also activates expression of *comEC* and *comEA* via the LuxR-family transcription factor QstR [11,43,44]. The regulatory mechanisms are unknown and additional factors must be involved; QstR does not bind the *comEA* regulatory region [43]. At low cell density, we propose that AphA plays an essential role and directly represses competence (Table 1, Fig 6). Like other key factors, AphA targets *tfoX* expression (Fig 3). Surprisingly, the ability of AphA to repress competence is apparent even when *dns* is present (Fig 5). Indeed, for bacteria locked at low cell density (i.e. when HapR is poorly expressed), deletion of *dns* had a comparatively small effect observed only when *aphA* was absent (Fig 5). This indicates that *aphA*-dependent repression of *tfoX*, rather than loss of *dns* repression by HapR, plays the dominant role in limiting natural transformation at low cell

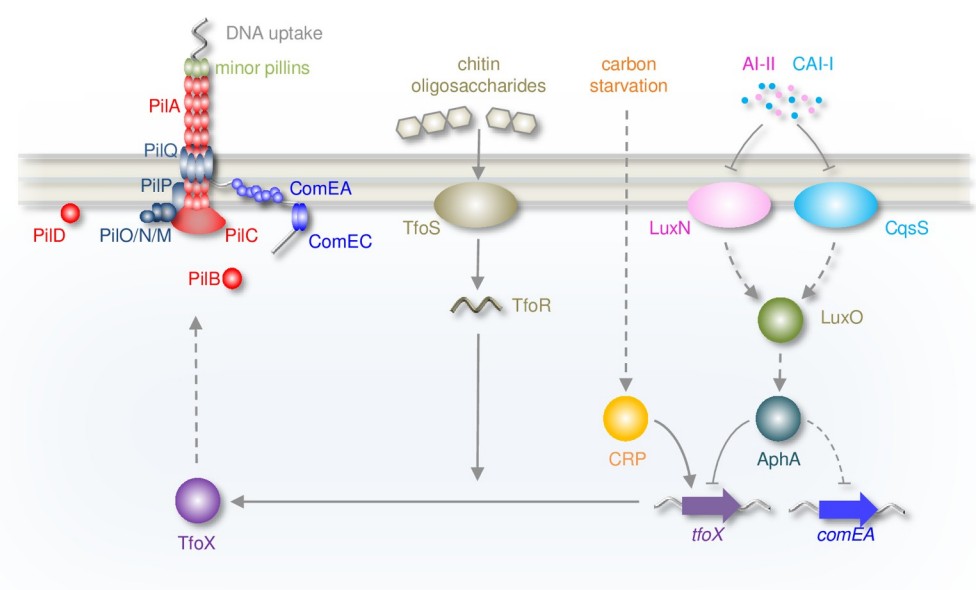

**Fig 6. Model for AphA-dependent regulation of natural competence.** The diagram illustrates regulatory pathways controlling the induction of natural competence. Solid and dashed arrows represent direct and indirect activation respectively. Barred lines indicate repression. Proteins are shown as ovals and spheres whilst genes are shown as arrows. Nucleic acids are shown as grey (DNA) or peach (RNA) wavy lines. The DNA uptake machinery comprises PilABCD, PilQPONM, ComEA, ComEC, VC0857 and VC0858 (encoding the minor pilins). Repression of competence by AphA primarily occurs at the level of *tfoX* repression. A small number of factors and intermediary steps have been omitted for clarity.

density. Near maximal competence can be induced at low population density if *aphA* is deleted (Fig 5). Additionally, artificial expression of AphA blocks competence at high-cell density, when HapR is abundant. Taken together, this suggests that AphA is a key link between competence and population density.

Mechanistically, AphA blocks competence induction by preventing the activator CRP binding to the *tfoX* promoter (Fig 3). The ability of AphA to displace CRP, even with the latter in excess, is of key importance. This ensures that *tfoX* expression cannot be switched on by carbon starvation and chitin metabolism alone; the bacterial population must also reach an appropriate level. Our observations concerning AphA and CRP at the *tfoX* promoter are reminiscent of the *tcpPH* regulatory region. This locus also has overlapping sites for the two regulators that act antagonistically [45]. Opposing regulation was also observed at the *cqsS* promoter, although the regulators targeted distinct sites on the DNA (Fig 4). We speculate that antagonistic control of promoters by AphA and CRP is a common regulatory strategy in *V. cholerae*.

Unexpectedly, we found that AphA could interact with single DNA sites or co-operatively bind pairs of targets. In the examples tested here, binding a single site required the sequence to closely match the consensus for AphA binding (Fig 2). More divergent AphA sites could function in unison by co-operatively binding the regulator (S2 Fig). These different configurations of DNA binding may explain why three previous studies each proposed a slightly different AphA binding consensus [24,26,27]. Unusually, none of these sequences were palindromic. We suggest that AphA preferentially binds to the inverted repeat sequence 5'-ATGCAA-N$_4$-TTGCAT-3' (Fig 1D). Consistent with this, structures of PadR family regulators demonstrate DNA binding as a dimer with two-fold symmetry [34]. Confusion likely arose previously because the sequence 5'-TGCA-3' is embedded as a direct repeat within the larger motif identified here (Fig 1D). Furthermore, a paucity of known AphA binding sites hindered prior studies.

Surprisingly, transcriptome analysis found only six differentially regulated operons in *V. cholerae* cells lacking *aphA*. Similar studies in *V. harveyi* identified hundreds of genes [22,25,46]. We speculate that these discrepancies result from the growth conditions used. This may also explain why control of the competence regulon was not identified. In particular, transcription of *tfoX* and *cqsS* requires CRP, but transcriptome analysis used rich media that triggers catabolite repression [46]. Indirectly, this would impact VC0857 that is induced by TfoX. Our data are consistent with transcriptome profiling of cells treated with quorum sensing molecules. Hence, VC0857, *tfoX* and *cqsS* are all activated by auto-inducers that block expression of AphA (Table 1) [31]. Similarly, like Rutherford and co-workers, we note that AphA frequently targeted genes involved in cell envelope physiology [22]. In hindsight, this is not surprising since the TCP is itself membrane associated [47]. Importantly, our ChIP-seq analysis did detect binding of AphA at the *tcpPH* locus, albeit at low levels (S5 Fig). We were also able to detect AphA binding at its own promoter, consistent with previous reports of auto-regulation (Table 1) [22]. However, we did not detect AphA binding upstream of *pva* or *alsR*, the only other known targets [46,48].

In summary, our work better defines DNA binding by AphA and expands the direct regulon by >10-fold. Of particular interest are genes repressed by AphA that play key roles in the control of natural competence. Hence, as well as inducing the pathogenicity cascade at low population densities, AphA plays a key role by repressing genes utilised in the aquatic environment. We caution that our definition of the AphA regulon is unlikely to be complete. However, our work provides a solid basis for understanding changes in gene expression caused by transition of *V. cholerae* between the environmental niche and human host. In particular, we explain how competence can be controlled in this regard.

## Materials and methods

### Strains, plasmids and oligonucleotides

Standard procedures for cell culture and storage were used throughout. Strains were constructed using the approach of Dalia *et al*. [12]. Full descriptions of materials used are in S1 Table. Derivatives of pRW50T were transferred from *E. coli* DH5α into *V. cholerae* by tripartite mating. Overnight cultures were washed twice using 0.9% (*w/v*) NaCl, resuspended in LB, mixed in a 1:1:2 ratio of donor:recipient:helper, then spotted on non-selective LB plates. After overnight incubation at 30 °C, cells were resuspended in 0.9% NaCl and plated on TCBS agar containing 100 µg/ml streptomycin and 5 µg/ml tetracycline. After overnight incubation at 37 °C, colonies were re-streaked on LB agar containing 100 µg/ml streptomycin and 5 µg/ml tetracycline. Conjugants were confirmed by PCR.

### Chromatin immunoprecipitation

ChIP-seq experiments were done as described in Haycocks *et al*. [49]. Briefly, *V. cholerae* E7946 was transformed with plasmid pAMCF*aphA* or pAMNF*aphA*. These encode AphA with a C- or N-terminal 3xFLAG epitope respectively. The N-terminally tagged AphA was unable to bind DNA in ChIP-seq experiments and so served as a useful control. Note that levels of AphA produced from plasmid pAMCF*aphA* and the native chromosomal locus were indistinguishable (S3 Fig). Cultures were incubated aerobically to mid-log phase in LB media at 37 °C. Cells were cross-linked with 1% (*v/v*) formaldehyde, washed, treated with lysozyme, and sonicated. The AphA-DNA complexes were immunoprecipitated with an anti-FLAG antibody (Sigma) and Protein A sepharose beads. Immunoprecipitated DNA was blunt-ended, A-tailed, and ligated to barcoded adaptors before elution and de-crosslinking. ChIP-seq libraries were then amplified by PCR and purified. Library quality was assessed using an Agilent

Tapestation 4200 instrument and quantity determined by qPCR using an NEBnext library quantification kit (NEB).

## Illumina sequencing and data analysis

Libraries were sequenced as previously described [50]. Each library was diluted to a concentration of 2 nM, before pooling and denaturation. Sequencing was done using an Illumina MiSeq instrument. Fastq files were deposited in Array Express (accession number E-MTAB-7953). Individual sequence reads were mapped against the *Vibrio cholerae* N16961 genome (Genbank accession numbers CP024162.1 and CP024163.1) using BWA (Burroughs-Wheeler Aligner) [51]. This facilitated comparison with other studies. Resulting Sequence Alignment Map (SAM) files were converted to Binary Alignment Map (BAM) files using the SAM-to-BAM tool [52,53]. Coverage per base was calculated using multiBAMsummary [54], and R was used to normalise each data set to the same overall read depth for each chromosome. To visualise the AphA binding profile coverage depth was plotted as a graph against genome features in Artemis. The graph window size was set to 100 bp and peaks with a coverage score of $\geq 10$ over 300 consecutive bases were selected. The centre of the region passing the cut off was set as the peak centre. Next, 250 bp DNA sequences from each peak centre were collated. To identify DNA motifs associated with peak sequences we used MEME [55]. We scanned for motifs between 12 and 26 bp in length that occurred once per given sequence on the given DNA strand.

## Natural transformation

Chitin-induced transformation assays of *V. cholerae* were done as described by using shrimp chitin flakes to induce competence (Sigma) [12]. Briefly, cells were grown to an $OD_{600}$ of ~1 in LB. Cells in 1 ml of culture were recovered by centrifugation and washed twice with 1 ml 0.7% (w/*v*) Instant Ocean (Aquarium Systems). Cells were diluted 10-fold with 0.7% Instant Ocean and 1 ml added to 10 mg of sterile chitin in a 2 ml Eppendorf tube. After incubation at 30 °C for 24 hours 200 ng of transforming DNA was added. Following a further 5 hours incubation cells were recovered in LB for 1–2 hours at 37 °C with shaking. Cells were then plated for quantitative culture on selective media (to quantify the number of transformants) and non-selective media (to quantify the total viable counts). The transformation frequency is defined as the number of transformants divided by the total viable count in each reaction.

## Protein purification and western blotting

*V. cholerae* CRP was purified using cAMP-agarose as previously described [56]. RNA polymerase was purified from *V. cholerae* N16961 using a protocol based on the method of Burgess and Jendrisak as previously described [56,57]. *V. cholerae* $\sigma^{70}$ was purified by affinity chromatography as previously described [56]. To facilitate overexpression *aphA* was cloned in pET21a. The resulting plasmid was used to transform *E. coli* T7 Express cells. All colonies resulting from a single transformation were pooled and used to inoculate 500 ml of LB supplemented with 100 μg/ml ampicillin. Overexpression of C-terminally $His_6$ tagged AphA was induced with 1 mM IPTG for 3 hours. Cells were recovered by sonication and resuspended in 20 mM Tris-HCl pH 7.5, 1 mM EDTA pH 8.0, 10 mM NaCl and 1 mM PMSF. After cell lysis by sonication cell debris was removed by centrifugation and the lysate was passed through a His-Trap column (GE Healthcare). Proteins were eluted with a gradient of buffer containing 20 mM Tris-HCl pH 7.5, 1 mM EDTA pH 8.0, 10 mM NaCl and 500 mM imidazole. Fractions with AphA were identified by SDS-PAGE, pooled and concentrated. For long-term storage at

-20 °C, glycerol was added to a final concentration of 50% *(v/v)*. Western blots were done as described by Lamberte *et al.* [58].

## Electrophoretic mobility shift assays (EMSAs)

DNA fragments for EMSA experiments were generated by PCR as previously described [59]. PCR products were cut using *Eco*RI and *Hin*dIII (NEB). End-labelling was done using $\gamma^{32}$-ATP and T4 polynucleotide kinase (NEB). Radiolabelled fragments were incubated with purified proteins in buffer containing 40 mM Tris acetate pH 7.9, 50 mM KCl, 5 mM $MgCl_2$, 500 µM DTT and 12.5 µg/ml Herring Sperm DNA for 15 minutes at 37 °C. Protein-DNA complexes were separated by electrophoresis using a 7.5% non-denaturing polyacrylamide gel. Subsequently, dried gels were exposed to a Biorad phophorscreen that was scanned using a Biorad Personal Molecular Imager. Full gel images are shown in S6 Fig.

## DNAse I and $KMnO_4$ footprinting

DNA fragments were excised from pSR using *AatII-Hin*dIII. After end-labelling using $\gamma^{32}$-ATP and T4 PNK (NEB), footprints were done as previously described in buffer containing 40 mM Tris acetate pH 7.9, 50 mM KCl, 5 mM $MgCl_2$, 500 µM DTT and 12.5 µg/ml Herring Sperm DNA [60,61]. Resulting DNA fragments were analysed on a 6% denaturing gel. Subsequently, dried gels were exposed to a Biorad phophorscreen that was scanned using a Biorad Personal Molecular Imager. Full gel images are shown in S6 Fig.

## In vitro *transcription assays*

We used the protocol of Kolb *et al.* [62] as described by Savery *et al.* [63]. Reactions contained different combinations of 16 µg/ ml supercoiled pSR template, *V. cholerae* RNA polymerase $\sigma^{70}$ holoenzyme, AphA and CRP. In experiments where CRP was used, CRP was pre-incubated with cAMP 37 °C prior to addition. The reaction buffer was 40 mM Tris pH 7.9, 5 mM $MgCl_2$, 500 µM DTT, 50 mM KCl, 100 µg/ ml BSA, 200 µM ATP/GTP/CTP, 10 µM UTP and 5 µCi $\alpha$-$P^{32}$-UTP. If required, AphA and CRP were added to reactions for 10 minutes at 37 °C before the addition of 0.4 µM RNA polymerase, for a further 10 minutes. Transcripts were analysed on a 6% denaturing polyacrylamide gel. The dried gel was exposed to a Biorad Phosphorscreen, which was scanned using a Biorad Personal Molecular Imager. Full gel images are shown in S6 Fig.

## β-galactosidase assays

*V. cholerae* harbouring pRW50T were grown to mid-log phase ($OD_{650}$ of ~1) in LB or M9 minimal media supplemented with 0.3% fructose unless stated in figure legends. Cells were lysed using 1% sodium deoxycholate and toluene, and assays carried out as previously described using the Miller method [56,64]. For cells containing plasmid pMMB-*tfoX* 1 mM IPTG was present.

## Microscopy

Strain $P_{tac}$-*tfoX* Δ*luxO* containing pAMCF*aphA* or pAMNF*aphA* was grown in continually rolling containers at 30 ºC to late-log phase in LB Miller broth supplemented with 50 µg/mL Kanamycin, 100 µM IPTG, 20 mM $MgCl_2$ and 10 mM $CaCl_2$. Around $10^8$ colony-forming units (c.f.u.) were collected by centrifugation (18,000 x *g* for 1 minute) and resuspended in 0.7% Instant Ocean. Cells were then labelled with 25 µg/ml AlexaFluor488 maleimide (AF488-mal) in the dark for 15 minutes at room temperature. Cells were then washed twice

with 0.7% Instant Ocean by sequential centrifugation and resuspension. Cells were imaged by time-lapse microscopy every 2 seconds for 2 minutes to monitor pilus production. The number of cells that made at least one pilus within the 2-minute window and the total number of cells were manually counted. To examine DNA internalisation, approximately $10^8$ c.f.u. of late-log culture were diluted 4-fold with Instant Ocean. The cells were then incubated with or without 100 ng MFP488-labelled DNA at room temperature in the dark. After 30 minutes, 10 units of DNase I (NEB) was added to all reactions and incubated for 2 minutes to degrade any remaining extracellular DNA. Cells were then washed twice with 0.7% Instant Ocean by sequential centrifugation and resuspension. Static images of cells were taken and the number of cells with a DNA-uptake event, indicated by MFP-488 DNA foci, compared to the total number of cells in a field of view, was manually counted. For all microscopy experiments, samples were placed under 0.2% Gelzan (Sigma) pads made with Instant Ocean medium. We used a Nikon Ti-2 microscope with a Plan Apo x60 objective, GFP and dsRed filter cubes and a Hamamatsu ORCAFlash4.0 camera. Image collection and analysis used Nikon NIS Elements imaging software and Image J.

## Supporting information

**S1 Fig. Binding of AphA to different DNA fragments *in vitro*.** The gel images show results of electrophoretic mobility shift assay using DNA fragments corresponding to the indicated intergenic DNA regions. The *lacZ* promoter region used was from *E. coli* whilst other DNA fragments were derived from *V. cholerae*.
(PDF)

**S2 Fig. Binding of AphA to the *tfoX* and *cqsS* regulatory regions.** a. Binding of AphA to the *tfoX* regulatory region and derivatives. The gel shows the result of a DNase I footprint using DNA fragments containing the *tfoX* regulatory region or derivatives with point mutations in individual AphA sites. Lanes 1–4 show results for the wild type DNA fragment. Lanes 5–7 and 8–10 show results with AphA I or AphA II mutated respectively. A Maxim-Gilbert 'G+A' ladder has been used to calibrate the gel (lane 1). Lanes 2, 5 and 8 show the pattern of DNase I cleavage in the absence of AphA. Where added, AphA was present at concentrations of 0.5 or 1.0 μM.
b. Binding of AphA to the *cqsS* regulatory region and derivatives. The gel shows the result of a DNase I footprint using DNA fragments containing the *cqsS* regulatory region or derivatives with point mutations in individual AphA sites. Lanes 1–4 show results for the wild type DNA fragment. Lanes 5–7 and 8–10 show results with AphA I or AphA II mutated respectively. A Maxim-Gilbert 'G+A' ladder has been used to calibrate the gel (lane 1). Lanes 2, 5 and 8 show the pattern of DNase I cleavage in the absence of AphA. Where added, AphA was present at concentrations of 2.0 or 3.0 μM.
(PDF)

**S3 Fig. Levels of chromosomal and plasmid encoded AphA are indistinguishable.** a. Image of a western blot to compare levels of AphA3xFLAG generated from the native chromosomal locus for *aphA* (lanes 2–4) and from plasmid pAMCF*aphA* (lanes 5–7). Individual lanes are replicates and equal amounts of total cellular protein were loaded in each lane. Cultures were harvested at an $OD_{650}$ of 0.6.
b. Quantification of relative AphA3xFLAG levels. The AphA3xFLAG band intensity for each lane in panel a was determined using Quantity One software. The average band intensity is shown and error bars represent the standard deviation of the three experimental replicates shown in panel a. P = 0.299 was calculated using a two-tailed Student's t-test and there was no

significant difference in AphA levels when comparing the two expression methods (NS).
(PDF)

**S4 Fig. AphA does not bind to the *comEA* regulatory region *in vivo*.** Genes are shown as block arrows. ChIP-seq coverage plots are shown for individual experimental replicates. Data for AphA are in teal and control profiles are grey. Signals above or below the horizontal line correspond to reads mapping to the top or bottom strand respectively.
(PDF)

**S5 Fig. AphA binds to the *tcpPH* regulatory region *in vivo*.** a. ChIP-seq data for AphA binding at the *tcpPH* locus. Genes are shown as block arrows. ChIP-seq coverage plots are shown for individual experimental replicates. Data for AphA are in teal and control profiles are grey. Signals above or below the horizontal line correspond to reads mapping to the top or bottom strand respectively.
b. Sequence of the *tcpPH* intergenic region. Coding DNA is in blue. Promoter elements are underlined and labelled. The transcription start site is shown as a bent arrow. Distances are with respect to the *tcpPH* transcription start site. The centre of the AphA ChIP-seq peak is denoted by a teal asterisk. The known AphA binding site is boxed.
(PDF)

**S6 Fig. Raw gel images.** Complete raw gel images are shown and subsections of images used for figures are indicated.
(PDF)

**S1 Movie. Pilus dynamics in the presence of AphA with an N-terminal 3xFLAG fusion.** To monitor pilus dynamics, we utilised *V. cholerae* encoding PilA$^{S67C}$. The cysteine substitution facilitates *in vivo* labelling of PilA with the fluorescent dye AF488-mal. The strain used is P$_{tac}$-*tfoX*, Δ*luxO*, *comEA*-mCherry and harbours the pAMNF*aphA* vector.
(AVI)

**S2 Movie. Pilus dynamics in the presence of AphA with a C-terminal 3xFLAG fusion.** To monitor pilus dynamics, we utilised *V. cholerae* encoding PilA$^{S67C}$. The cysteine substitution facilitates *in vivo* labelling of PilA with the fluorescent dye AF488-mal. The strain used is P$_{tac}$-*tfoX*, Δ*luxO*, *comEA*-mCherry and harbours the pAMCF*aphA* vector.
(AVI)

**S1 Table. Strains, plasmids and oligonucleotides.**
(DOCX)

## Acknowledgments

We would like to thank Joseph Wade for providing feedback on the manuscript prior to submission.

## Author Contributions

**Conceptualization:** Lucas M. Walker, David C. Grainger.

**Data curation:** James R. J. Haycocks, Gemma Z. L. Warren, Jennifer L. Chlebek, Ankur B. Dalia, David C. Grainger.

**Formal analysis:** Gemma Z. L. Warren, Jennifer L. Chlebek, Ankur B. Dalia, David C. Grainger.

**Funding acquisition:** Ankur B. Dalia, David C. Grainger.

**Investigation:** James R. J. Haycocks, Gemma Z. L. Warren, Jennifer L. Chlebek, Triana N. Dalia, Ankur B. Dalia, David C. Grainger.

**Methodology:** James R. J. Haycocks, Jennifer L. Chlebek, Triana N. Dalia, Ankur B. Dalia, David C. Grainger.

**Project administration:** Triana N. Dalia, David C. Grainger.

**Resources:** David C. Grainger.

**Software:** James R. J. Haycocks, David C. Grainger.

**Supervision:** James R. J. Haycocks, Ankur B. Dalia, David C. Grainger.

**Validation:** James R. J. Haycocks, Ankur B. Dalia, David C. Grainger.

**Visualization:** James R. J. Haycocks, Jennifer L. Chlebek, Ankur B. Dalia, David C. Grainger.

**Writing – original draft:** James R. J. Haycocks, Lucas M. Walker, David C. Grainger.

**Writing – review & editing:** James R. J. Haycocks, Gemma Z. L. Warren, Lucas M. Walker, Jennifer L. Chlebek, Ankur B. Dalia, David C. Grainger.

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
