## [Decision Letter · Decision Letter 0]

17 Sep 2019

 Dear Dave,

Thank you very much for submitting your Research Article entitled 'The quorum sensing transcription factor AphA directly regulates natural competence in Vibrio cholerae' to PLOS Genetics. Your manuscript was fully evaluated by three independent peer reviewers as well as by myself, given my deep knowledge of the topic. Please note that, as you provided a rebuttal letter addressing your initial submission to eLife, the experts and I also took this letter into consideration.    As you can see from the comments that are attached below, the experts felt that your study is well executed and that the biochemical and* in vitro* work is particular rigorous and strong. However, consistent with the comments provided by the reviewers at *eLife*, some of the current experts commented once again on the fact that the significance of the role of AphA in the natural competence process is still not entirely clear. Indeed, the title, which implies a major contribution of AphA to competence regulation, is not convincing given that the competence enhancement of an *ahpA *mutant only becomes obvious after the deletion of *dns*. I fully agree with this judgement, as the nuclease Dns is a major inhibitor of natural transformation in Vibrio cholerae. For this reason, *dns *is partially repressed by the quorum sensing regulator HapR but fully repressed via the TfoX-QstR pathway. The latter pathway is not taken into consideration in your experimental setup and the data therefore do not fully underpin your claim that AphA is a major player in the competence process in WT strains. Indeed, the exquisite timing between *dns *repression and the production of the DNA-uptake machinery is of prime importance for natural transformation and taking Dns out of the equation isn't ideal.   Given these remaining problems, I can offer you a minor revision which would, however, require significant textual changes. Such textual changes should aim at properly stating what your study shows and where the limitations are, including revision of the title (e.g., "The QS transcription factor AphA represses competence in Vibrio cholerae if expressed at high cell density") and an amendment of the abstract (e.g., “…we demonstrate that AphA also intervenes in cells that are locked in a low cell density state.” and removal of the “dual regulation”, as this gives too much emphasis on the AphA repression, despite the fact that this repression would not occur under normal high cell density competence-inducing conditions - consistent with the absence of a phenotype of the *ahpA *mutant at HCD).

In summary, we ask you to modify the manuscript according to the reviewers' and editor's recommendations before we can consider your manuscript for acceptance. Your revisions should also address the specific points made by each reviewer and the editor (see below).

[LINK]

Yours sincerely,

Melanie 

Melanie Blokesch

Associate Editor

PLOS Genetics

Josep Casadesús

Section Editor: Prokaryotic Genetics

PLOS Genetics

Reviewer's Responses to Questions

**Comments to the Authors:**

Editor's comments: 

These minor issues should also be addressed in the revised manuscript: 1) I highly encourage you to revise some citations (e.g., line 28f: that chitin induces TfoX was not only shown by Yamamoto et al., 2010, 2011, 2014 but long before that by the Schoolnik lab - Meibom et al 2004; Meibom et al., 2005  - with the latter study also highlighting that TfoX induction is sufficient to foster competence induction under rich medium conditions.  2) In line 84ff you addressed the genes VC0857 and VC0858. It was shown in the past that deletion of VC0857 had only a minor effect on transformation under chitin-inducing conditions (~1 log; Seitz and Blokesch, PNAS 2013; Table S4). It is therefore not surprising that repression of this gene by artificially induction of AphA at HCD doesn’t do much (line 271). This should be discussed. 3) Subtitle line 224 needs amendment such as “under nuclease-deficient conditions” .  4) Line 247 referring to Fig. 5a: The graph shows transformation frequencies (e.g., the sum of uptake of non-degraded DNA over the OM, DNA translocation over the IM, recombination, and resistance gene expression) and not DNA uptake. Distinguishing between those two things is important, as the change might not be fully caused by DNA uptake per se. 5) Discussion section line 298 (and elsewhere): “We propose that AphA directly couples competence to population density signals” => this has been know to occur via HapR for many years (e.g., since Meibom et al., 2005) and is not a new proposition. Please revise to be more precise on what exactly AphA’s repression contributes to competence regulation and how this differs from the effect exerted by HapR.  6) Please note that “data not shown” is not acceptable by PloS Genetics. The respective data should be provided (e.g., line 115).

Reviewer #1: In this revised submission, Haycocks et al. perform ChIP-seq to identify genes that are directly regulated by the V. cholerae low cell density quorum-sensing regulator AphA. The authors define multiple sites in the V. cholerae genome to which AphA binds, most notably the gene encoding the master regulator of competence, tfoX. As noted in my writeup about the earlier submission, all of the biochemical and genetic assays are extremely sound and convincing.

In the previous submission, the major issue concerned the competence aspects of the manuscript. In particular, the title indicated that AphA regulated competence, however, the experiments were not provided to convincingly underpin that claim: AphA was shown to affect competence only when expressed ectopically, and at cell density states in which native AphA expression is known be highly repressed (high cell density). In the resubmitted version, the authors have dealt with this issue by measuring competence in a strain harboring the well-described, phosphomimetic, LuxOD47E allele, in which the cells are locked into a low cell density state. Thus, the authors are now probing AphA function under a condition in which AphA is relevant. Using this strategy, the authors show that deletion of aphA leads to a dramatic enhancement in competence.

One issue with the new experiments is that the gene encoding a nuclease, dns, also had to be deleted from the locked strain to show the effect of AphA. As the authors note: “It was also necessary to delete dns; the gene encodes an endonuclease expressed at low cell density to degrade any DNA obtained by natural transformation” (line 230-231). The fact that Dns is required “to degrade any DNA obtained by natural transformation” leads the reader to once more question whether the effect of AphA on competence is particularly relevant. In this newly submitted version, the authors successfully deal with the issue of the previous version by studying native levels of AphA, but their new findings show that Dns prevents natural transformation, which renders repression of the competence machinery by AphA much less significant, or at best, much more complicated than anticipated. There may well be situations in which AphA is present and Dns absent, however, it is worth noting that the authors did not find one here.

Reviewer #2: This manuscript by Haycocks et. al. describes new insight into the master low-cell density regulator AphA in Vibrio cholerae. The authors make a number of important conclusions including mapping the DNA binding sites of AphA on the V. cholerae chromosome, exploring the mechanism of AphA gene regulation in detail at three different binding sites, and demonstrating that AphA is a negative regulator of competence in V. cholerae. I found this work was a very nice combination of system wide analysis with ChIP-seq and more mechanistic studies such as DNAse protection assays, EMSAs, and in vitro transcription. I think the conclusions are important, significantly increase our understanding of global regulatory pathways in V. cholerae, and their data support their conclusions. Thus, I only have a few minor comments about the manuscript.

1. Herzog et. al., 2019, NAR should be cited and described in the introduction, and the results from thie work should be compared to what they found in the discussion.

2. Lines 63-64. This region should be indicated on Chromosome II.

3. Table 1: It would be helpful to include on Table 1 the gene names for any of these genes in which a function has been described or a gene name assigned along with the “VC” gene designation.

4. Line 97: It might be worth explaining that the transcriptional start sites were previously identified using a transcriptomics approach, otherwise, readers might think they were identified in this study and be confused about the lack of explanation or description about methods for doing this.

5. Line 124: How does Table 1 show that VC0858 is unresponsive to AIs?

6. Fig. S2 is referred to extensively in the manuscript, even in the discussion section, so I think it should be moved from the supplemental data to the main manuscript.

7. Figs. 4e, 4f: I think it is commendable that the authors confirmed their in vitro transcription results with the KMnO4 assay and speaks to the thoroughness of the data.

8. It has been shown that the translation start site of LuxO was initially incorrectly identified 14 codons to the 3’ of the actual translation start site (see methods section PMID: 22295878). Thus, the aspartate that is phosphorylated in LuxO is not actually at position 47, but rather at position 61. So, the LuxO mutant used should really be described as D61E, not D47E.

Reviewer #3: In this manuscript, the authors use ChIP-seq to identify genes that are regulated by the transcription factor AphA. Using this technique, they identify several genes that were not previously known to be AphA regulated, including three genes involved in natural competence, tfoX, cqsS, and VC0857. The authors identified the DNA motif required for AphA binding, and perform detailed analyses of this binding at those three promoters. Finally, they demonstrate that the direct repression of tfoX expression by AphA results in a reduction in bacterial competence. Overall, this is a detailed and well-controlled study. This work was previously reviewed for another journal, and clearly has been updated/improved based on the previous reviewers’ suggestions. There are fairly minor aspects of the manuscript that could still be improved.

Specific points:

1. The figure legends are extensive and somewhat cumbersome. Due to the fact that figures 2,3, and 4 follow the same format, it seems like they could be shortened in some way, or that some of the information could be moved to the materials and methods instead of using the legends.

2. While all of the figures are fairly dense and contain a lot of data, figure 1 is harder to navigate. Having panel “f” up in the middle of the figure is confusing, when the rest of the figure reads from top to bottom. I would suggest reworking the figure or text in some way that makes the order of the figure make more sense, or perhaps splitting the information into two figures.

3. In Table 1, it might be helpful to identify which regulated genes were identified from other studies (and cite those studies) so the readers know what was previously known and what is new. It might also help to include gene names in addition to VC numbers.

4. Figure 5f is not referenced in the text. I believe that it should be referenced on line 290.

**Have all data underlying the figures and results presented in the manuscript been provided?**

Reviewer #1: Yes

Reviewer #2: Yes

Reviewer #3: Yes

PLOS authors have the option to publish the peer review history of their article (what does this mean?). If published, this will include your full peer review and any attached files.

Reviewer #1: No

Reviewer #2: No

Reviewer #3: No

---

## [Editor Report · Decision Letter 1]

16 Oct 2019

Dear Dave,

We are pleased to inform you that your manuscript entitled "The quorum sensing transcription factor AphA directly regulates natural competence in Vibrio cholerae" has been editorially accepted for publication in PLOS Genetics. Congratulations!

**Just one last point: please ensure that your tables lists all strains and material that were used in this study so that researchers can request them accordingly. In its current version, it looks like none of the strains that were used for the first and second revision were added to the final manuscript files (e.g., nuclease-deficient strains and LCD-locked strain etc) and neither were the oligonucleotides that were used to construct these strains. **

Yours sincerely,

Melanie

Melanie Blokesch

Associate Editor

PLOS Genetics

Josep Casadesús

Section Editor: Prokaryotic Genetics

PLOS Genetics

Comments from the reviewers (if applicable):

**Data Deposition**

http://datadryad.org/submit?journalID=pgenetics&manu=PGENETICS-D-19-01332R1

**Press Queries**

---

## [Editor Report · Acceptance letter]

24 Oct 2019

PGENETICS-D-19-01332R1 

The quorum sensing transcription factor AphA directly regulates natural competence in *Vibrio cholerae*

Dear Dr Grainger, 

We are pleased to inform you that your manuscript entitled "The quorum sensing transcription factor AphA directly regulates natural competence in *Vibrio cholerae*" has been formally accepted for publication in PLOS Genetics! Your manuscript is now with our production department and you will be notified of the publication date in due course.

With kind regards,

Nicholas White

PLOS Genetics

On behalf of:
